# Senescence and Aging: Does It Impact Cancer Immunotherapies?

**DOI:** 10.3390/cells10071568

**Published:** 2021-06-22

**Authors:** Damien Maggiorani, Christian Beauséjour

**Affiliations:** 1Centre de Recherche du CHU Ste-Justine, Montréal, QC H3T 1C5, Canada; damien.maggiorani@gmail.com; 2Département de Pharmacologie et Physiologie, Faculté de Médecine, Université de Montréal, Montréal, QC H3T 1J4, Canada

**Keywords:** immune checkpoint inhibitors, aging, senescence, cancer, tumor

## Abstract

Cancer incidence increases drastically with age. Of the many possible reasons for this, there is the accumulation of senescent cells in tissues and the loss of function and proliferation potential of immune cells, often referred to as immuno-senescence. Immune checkpoint inhibitors (ICI), by invigorating immune cells, have the potential to be a game-changers in the treatment of cancer. Yet, the variability in the efficacy of ICI across patients and cancer types suggests that several factors influence the success of such inhibitors. There is currently a lack of clinical studies measuring the impact of aging and senescence on ICI-based therapies. Here, we review how cellular senescence and aging, either by directly altering the immune system fitness or indirectly through the modification of the tumor environment, may influence the cancer-immune response.

## 1. Introduction

Improved life conditions and medical innovations have led to an increased lifespan [1]. While the risk of developing cancer rises in patients over 60 years old, cancer incidence actually decreases at the age of 85 for reasons that are not fully understood but that likely involve the decline in cells’ proliferative potential [2,3,4]. The accumulation of senescent cells plays an important role in the risk of developing several age-related diseases, including cancer [5,6]. However, it is still unknown whether cellular senescence affects the efficacy of anticancer treatments and, in particular, the ability of cancer cells to evade the immune system [7]. This question is increasingly important with the emergence of promising new immunotherapy treatments. In this review, we highlight recent observations suggesting that aging and senescence can alter immune cell fitness and, ultimately, have an impact on the efficacy of cancer treatments.

## 2. Senescence: An Overview

### 2.1. Senescence Origin and Markers

The senescence phenotype is characterized by an irreversible cell cycle arrest and was first described by Moorhead and Hayflick in 1961 [8]. Dimri and Campisi then identified the first marker, which they named the senescence-associated-β galactosidase or SA-β-gal [9]. Since then, the field has widely evolved, and several other characteristics describing senescent cells have been identified, although none can be referred to as a universal marker [10,11]. Most often, it is best to identify a senescent cell through a combination of the following features: SA-β-gal, expression of a cyclin-dependent kinase inhibitor (CDKi) such as p16 and/or p21 [12], persistent telomeric DNA damage [13], the formation of a heterochromatin foci [14], a senescence-associated secretory phenotype (SASP) composed mostly by proinflammatory cytokines (IL-6 and IL-8), growths factors (VEGF and GM-CSF) and matrix-metalloproteinase (MMP) [15], and the accumulation of lipofuscin deposits [16]. The senescence phenotype can be engaged by a variety of stimuli such as telomeres erosion, leading to replicative exhaustion [17], exposure to genotoxic stressors (chemotherapy and ionizing radiation) [18], and the expression of a proto-oncogene, also known as oncogene-induced senescence (OIS) [19]. In addition, senescent cells have been shown to form through a bystander effect mediated by the SASP [20].

### 2.2. Senescent Cell Tracking In Vivo

The identification of senescent cells in vivo is further complicated by the heterogeneity and organ-specific temporal signatures of aging makers [21,22]. The expression of p16 has been widely used in vivo, since its expression increases in several organs in old mice [12,22]. Notably, there exists no reliable antibody capable of detecting the endogenous murine p16 protein, and this has led researchers to develop models to track p16-positive cells in vivo. Two of these mouse models, p16-INK-ATTAC [23] and p16-3MR [24], express fluorescent or luminescent reporter genes downstream the p16 promoter. Additionally, these mice express suicide genes (Caspase 8 and herpes simplex virus thymidine kinase, respectively), allowing for the specific killing of p16-positive cells. While these two models are extremely effective at eliminating p16-positive cells, the signal intensity from their reporter genes does not allow for the visualization of senescent cells at the single cell level in vivo. To our knowledge, the model developed by the team of Norman Sharpless, where a tandem dimer Tomato (tdTomato) reporter gene is knocked into the endogenous p16 locus in the presence of an enhancer, is the only model allowing the detection of p16-positive cells at the single-cell level [25]. Remarkably, the elimination of these cells in chronologically or accelerated models of aging decreases the onset of age-related diseases and cancer-related deaths [23,26]. It should be noted that, paradoxically, the presence of senescent cells is beneficial in instances such as wound healing [24] and embryogenesis [27].

The formation of senescent cells was shown to occur in cancer survivors exposed to genotoxic treatments and within tumors (referred as cancer treatment-induced senescence) [28,29,30,31]. As such, the development of senolytics, a new pharmacological class of drugs, allows for the specific clearance of senescent cells. Several senolytic drugs are currently being studied, such as Dasatinib + Quercetin (D + Q) [32] and navitoclax (ABT263) [33]. In mice, both drugs showed promising results in preventing many aged-related diseases, like metabolic disorders [34], osteoarthritis [35], dementia [36], pulmonary fibrosis [37,38], cardiac fibrosis/hypertrophy [39], and an immune imbalance [40]. Overall, senolytic approaches improve physical functions and, to a lesser extent, also increases the lifespan in mice [41]. Though not the subject of this review, several newly identified senolytic drugs are currently under investigation, such as fisetin [42], cardiac-glycosides [43,44], and CAR-T cells against murine senescent cells [45]. This is believed to be important if one seeks to eliminate all types of senescent cells. As we will discuss below, we believe that senolytic drugs could increase the efficacy of cancer immunotherapy treatments in several ways.

## 3. Aging in the Immunotherapy Era

Over the last 10 years, the development of immunotherapies has revolutionized cancer treatments [46]. Most immunotherapies are antibodies-based approaches aiming to boost immune cell functions against a tumor. Immunotherapy was developed against a negative regulator of the T-cell response, the cytotoxic T-lymphocyte antigen-4 receptor (CTLA-4), and identified as an immune checkpoint inhibitor (ICI) [47,48]. Then, another strategy organized against the immunosuppressive programmed cell death protein 1 (PD-1) or its ligand (PD-L1) was deployed [49,50]. However, the efficacy of ICI therapies is facing a high risk of cancer resistance or relapse [51]. Several studies are currently underway to validate the efficacy of other ICIs, including, but not limited to, LAG3, TIM3, the V-domain immunoglobulin suppressor of T-cell activation (VISTA), B7-H3, and TIGIT [52]. Interestingly, it was recently proposed that ICI can boost the abscopal effect [53,54]. The abscopal effect is defined by the immune-mediated resection of a secondary tumor (metastasis) distant from a primary tumor treated by radiotherapy (RT). The combination of RT with immunotherapies was shown to improve tumor regression in different cancer mouse models [55]. Clinically, the number of cases where an abscopal effect has been observed is increasing [56]. 

The importance of aging and senescence on the efficacy of ICI remains under-evaluated in preclinical studies. Indeed, most cancer patients are over 60 years old at the time of treatment, the equivalent of a 15–18-month-old mouse. Unfortunately, in most preclinical studies, mice are usually much younger, around 3 or 6 months old or the equivalent of 20-year-old patients [57]. We believe it is important to take into account the age factor in the experimental design to better predict the tumor treatment efficacy in humans. In this context, one publication showed that anti-CTLA-4 and anti-PD-L1 treatments were less efficient in reducing the growth of B16 melanoma cells in 19–26-month-old male and female mice [58]. Mechanistically, this study revealed a decreasing number of infiltrated and activated lymphocytes within the tumors of aged mice. However, the mechanisms explaining the aged-associated resistance to ICI remain mostly unknown and need to be investigated.

Analyzing the efficacy of ICI in clinic is difficult given that most cancer-treated patients should be considered immunologically old and, thus, without a true comparison with young patients [59]. Indeed, in all the meta-analyses published so far, the “young” group is defined as below <60–65 years old. We believe that patients younger than 60–65 years old cannot be defined as “young” and that immune fitness is already compromised, given the average age at which most cancer types emerge. Nonetheless, using this classification, it was interpreted that immunotherapy is not compromised with age, except for a possible loss of efficacy in patients over 75 years old [60,61,62]. On the opposite hand, a different analysis even showed a better survival rate in aged compared to young patients treated for metastatic melanoma [63,64]. The reasons for such discrepancies are difficult to explain, but we speculate that, above a certain age, when the tumor growth rate is reduced, it may indirectly help immune cells rejecting the tumor (see Section 4.7 Tumor Growth Rate and Aging)). Another possibility is that a greater number of immunosuppressive cells may be present in the tumors of younger patients, although this remains to be demonstrated [64].

## 4. Age-Associated Mechanisms Affecting the Efficacy of Immunotherapy

Here, we will discuss how we think aging and senescence can predominantly interfere with the efficacy of immunotherapies [65]. The efficacy of ICI is largely dependent on the tumor microenvironment (TME), which is composed by immune and nonimmune cells, such as cancer-associated fibroblasts (CAFs) and tumor-infiltrated leukocytes (TILs). CAFs shape and build up the tumor, whereas TILs, defined by a large family of lymphocytes and myeloid cells, have a double-edged role by either eliminating and/or feeding the tumor growth.

### 4.1. Accumulation of Senescent Cancer-Associated Fibroblasts

The identification of CAFs is challenging given their high heterogeneity and the lack of fibroblast-restricted markers [66]. The accumulation of CAFs in the TME is a negative prognosis, since they are a source of immunosuppressive cytokines such as IL-6, CXC-chemokine ligand (CXCL), and TGFβ [67]. CAFs are mainly derived from bone marrow progenitors [68] and the surrounding fat tissue [69]. Several components of the SAPS (i.e., the TGFβ family ligands, IL-6, IL-1, and CXCL-1) can contribute to the formation of CAFs [70,71,72,73]. Thus, the accumulation of senescent cells and the secretion of SASP within the TME may enhance the recruitment or the formation of CAFs and diminish the efficacy of ICI by consolidating an immunosuppressive environment [74].

How aging shapes the phenotype of CAFs is elusive and needs to be further investigated. Intrinsically, CAFs and senescent fibroblasts share common features, and the differences between them remain difficult to define. For example, CAFs and senescent fibroblasts express SA-β-gal and accumulate DNA damage [75]. The first demonstration that senescent fibroblasts, just like CAFs, have an impact on tumor growth came 20 years ago by Krtolica and Campisi, who showed that senescent human fibroblasts, when co-injected with pre-tumorigenic cell lines, boost tumor growth [76]. One limitation from this important study is that human cells had to be injected in nude mice and, thus, in the absence of most immune cells. Since then, the work by Ruhland and Stewart using immune-competent mice showed that murine senescent stromal cells can increase the recruitment of suppressive myeloid cells and promote tumor growth [77]. Similarly, senescent fibroblasts through the secretion of IL-6/IL-8 and amphiregulin were shown to favor the differentiation of monocytes into proinflammatory M2 macrophages and to drive the expression of PD-L1 in tumor cells, respectively [75,78]. In addition, CAFs stimulated with proinflammatory cytokines, such as IL-1α, IL-1β, and TNFα, tend to adopt a senescent state that favors tumor cell resistance to apoptosis and dissemination [73]. Whether human senescent fibroblasts, through their SASP, have positive or negative impacts on the tumor immune response is not clear at this point. To answer that question, one will need to use newly developed mouse models where tumor cells can be rejected in humanized mice [79]. While not the topic of this review, several studies have shown that cancer treatment-induced senescence has a major impact on the rejection of murine tumors by immune cells [80]. Interestingly, the systemic elimination of senescent cells was shown to decrease mammary tumor metastasis and chemotherapy-associated fatigue [81]. In summary, manipulating CAFs and senescent cells appears to be a promising strategy to decrease tumor growth. This will be better done when we learn more on how aging and senescence impact the accumulation of CAFs and shape their secretion profile.

### 4.2. Modification of Leukocytes with Age

TILs are mostly constituted of T cells and myeloid cells, and they play an important role in the tumor immune response. Yet, these cells have completely different functions within the TME. As we age, the number of circulating T cells tends to decrease, while there is no significant modification in the number of B cells [82]. Thymic involution, along with the architectural and functional alterations during aging, have been abundantly described to explain the reduction of circulating naïve T cells [83,84]. In addition, several publications showed a decline in the T-cell receptor (TCR) repertoire in aged naïve T cells, which can lead to an impaired immunity [84,85,86]. Alterations in bone marrow hematopoiesis during aging is associated with the accumulation of circulating myeloid cells, a process called myeloid skewing [40,87,88]. Lymphocytes (mostly CD8-positive T cells) are activated in secondary lymphoid organs (i.e., the spleen and lymph nodes) by specialized antigen-presenting cells (APC) and, then, are recruited within tumors. These cytotoxic CD8 lymphocytes are the main cells responsible for tumoral clearance, and their accumulation is associated with a good prognosis. However, activated T cells often become exhausted within tumors, a state characterized by the sustained expression of inhibitory receptors (i.e., PD-1 and CTLA-4) and associated with tumor immune resistance [89]. In mouse spleens, the number of highly exhausted T cells, which fail to proliferate after TCR activation and display enhanced IL-10 production, was shown to gradually increase with age [90]. Interestingly, the work by Sceneay et al. showed that blocking both PD-L1 and CTLA-4 was less efficient in old (>12 months old) compared to young mice in a triple-negative breast tumor model [91]. Mechanistically, it was shown that CD8 T cells in aged mice had a higher level of exhaustion and a diminished production of IFNγ. Interestingly, upon activation, T cells isolated from young mice were shown to exhibit a core transcriptional profile that was not as strongly activated and more heterogeneous in T cells collected from old mice [92]. This suggests that intrinsic transcriptional differences occur in T cells with age. Whether these differences were mediated by T-cell exhaustion or senescence, two processes regulated independently, was not investigated [93]. Overall, whether senescence and aging have an impact on the efficacy of ICI to reinvigorate exhausted TILs remains to be determined.

### 4.3. Are T Cells Really Senescent?

A big question in the field is whether T cells truly become senescent during aging and if it affects their functions [94]. The first evidence came when the expression of the senescence marker p16 was found increased in circulating T cells collected from both aged mice and humans [95,96]. However, these studies did not reveal the fraction of T cells expressing p16. This information came later using a reporter mouse model that allowed the tracking of p16-positive cells by flow cytometry, where it was shown that less than 1% of the circulating T cells expressed p16 in aged mice [25]. This was 5–10 times less than the frequency observed in cartilage and fat progenitor cells. On the opposite hand, a recent study showed that the frequency of human T cells expressing senescence markers could be as high as 60% in patients over 60 years old. These results were mostly based on the expression of SA-β-gal by flow cytometry, and whilst sorted positive cells expressed other markers of senescence, these were not quantified at the single-cell level [59]. Importantly, these senescent T cells were shown to be impaired in their ability to proliferate. The discrepancy in the estimated frequency of senescent T cells between these two studies can be explained by the difference between mice and humans and by the methodology used. Indeed, it is possible that the sensitivity of the reporter mouse model, based on Td-Tomato expression, although better than all other models available, underestimates the frequency of p16-expressing cells. Using an irradiation-induced premature aging mouse model, our group also demonstrated that T cells collected from the spleen had a higher p16 expression, but we found these cells to preserve their proliferative capacities in vitro. Only when T cells were mixed with full splenocytes, collected from an irradiated mouse, was their proliferation delayed, highlighting the importance of the splenic microenvironment [97]. 

Others have defined senescent T cells as a population expressing CD45RA and CD57 but deficient for CD28/CD27, a phenotype delineating T-effector memory (Temra) cells in humans [98]. This subpopulation of T cells was shown to increase with age; to overexpress several SASP-related genes (i.e., like IL-10, TNFα, and IFNγ); and to proliferate less than their double-negative counterparts [99,100,101]. These cells, often referred to as senescent-like, were also found to be associated with several aged-related pathologies, such as type 2 diabetes [102], acute heart failure [103], and acute proinflammatory situations like COVID-19-exposed patients [104]. In opposition, others demonstrated that human CD57-positive and CD57-negative T cells exhibit similar proliferation properties [105]. Altogether, it appears that T cells undergo cellular aging and senescence and that their functions are intrinsically or extrinsically diminished [94]. Keeping in mind that human T cells collected from a 60-year-old donor already displayed high levels of senescence markers, it is easy to speculate that the efficacy of ICI is likely already diminished at that age [59].

### 4.4. T-Regulatory Cells (Tregs)

Tregs are a subpopulation of immunosuppressive CD4 T cells that are often defined by the expression of the transcription factors Foxp3 and CD25 [106]. In aged mice, the number of Tregs increases, and these cells are more effective at suppressing the proliferation of effector T cells compared to Treg cells collected from young mice [107]. Moreover, Tregs isolated from aged mice overexpress IL-10 and strongly suppress the expression of CD86 in DC, an essential receptor for antigen presentation [107,108]. On the opposite hand, a recent manuscript showed that Tregs isolated from aged mice, expressing the p16 and p21 senescence markers, were less effective at decreasing T-cell activation compared to Tregs isolated from young mice [109]. We believe this discrepancy could be associated with the difficulty in identifying Tregs with current markers and the different methods used to assess T-cell activation. Interestingly, in a model of B16-resistant tumors, Treg depletion in 19–24-month-old mice partially improved the efficacy of the CTLA-4- but not PD-L1-blocking antibodies [58]. Hence, the impact of senescence and aging on Tregs remains much debated at this point.

### 4.5. Macrophages and Dendritic and Antigen-Processing Cells

Macrophages are myeloid specialized cells that, as part of the adaptive immune response, can phagocyte and present tumor antigens. During aging, macrophages are shown to express markers of senescence (i.e., p16 expression and SA-β-gal) [25]. Strikingly, these senescent macrophages exhibit a higher phagocytosis capacity and a decreased 5-Ethynyl-2′-deoxyuridine (EdU) incorporation. In mice, the injection of alginate beads into the peritoneal cavity leads to the accumulation of senescent-like macrophages positive for SA-β-gal and p16 [110,111]. In the TME, tumor-associated macrophages (TAM) promote neovascularization, tumor growth, and metastasis through cytokines (i.e., IL-6, IL-8, and IL-10) that clearly overlap with the SASP [112]. To our knowledge, only one study has demonstrated the detrimental role of TAM during aging: it was shown that TAMs pushed mesothelioma tumors to grow faster in aged vs. young mice [113]. Blocking these TAMs using F4/80 antibodies could restore the efficacy of an IL-2/anti-CD40-based immunotherapy in aged mice. 

Like macrophages, there is no clear evidence showing that dendritic cells (DC) can express classical senescence markers. However, DC isolated from aged mice were shown to exhibit an impaired capacity to cross-present antigens to CD8-positive T cells in vitro [114]. This may be linked to the observation that DC collected from aged mice have a defect in their antigen-processing machinery. In humans, a study showed that DC isolated from old donors (over 65 years old) secrete more SASP-related cytokines (IL-6 and TNFα) compared to the same cells isolated from younger donors (range 21–30 years old) [115]. Overall, there is little evidence describing and characterizing the senescence state of macrophages or DC. Hence, their impact on aging and on the efficacy of ICI treatments remains to be determined.

### 4.6. Neutrophils and MDSC

Similar to macrophages, tumor-associated neutrophils play an important role in the tumor immune response, and their accumulation is associated with a poor prognosis [116,117]. Neutrophils have an impact on many mechanisms, including promoting inflammation, extracellular matrix remodeling, angiogenesis, metastasis, and immunosuppression [116]. It is believed neutrophils release reactive oxygen species that can inhibit T-cell activation and express ligands that activate immune checkpoints [118,119,120]. Given their short half-lives, there is no evidence that neutrophils adopt a senescence phenotype. However, evidence suggests that neutrophils may be influenced by the SASP. Indeed, it is known that neutrophils can undergo cell death in the form of NETosis, where the nuclear DNA of neutrophils is extruded and projected into the surrounding tissue [121]. It was shown that the release of neutrophil extracellular traps (or NETs) within a tumor, by obstructing contact with immune cells, protect malignant cells [116,122]. A recent publication highlighted the roles of CXCR1 and CXCR2 in the production of NETs [123]. Interestingly, their ligands, CXCL8 (IL-8) and CXCL1 (GRO1/KC), respectively, are two major SASP factors [124]. Based on this information, we believe senescent cells within the TME, through their SASP, could induce the release of NETs and interfere with tumor immune clearance. 

Myeloid-derived suppressive cells (MDSC) represent a subpopulation of neutrophils. MDSC originate from the incomplete differentiation of common myeloid progenitors in the presence of inflammatory cytokines (i.e., IL6, TNFα, and MCP-1) within the TME. MDSC can be separated into two major subpopulations, monocytic-MDSC and granulocytic-MDSC, classified according to their origin from monocytic or granulocytic lineages, respectively [125,126]. Despite distinct origins, MDSC play an important role during tumorigenesis, as they have the ability to suppress T-cell activation by using several mechanisms, such as expressing Arginase1, CD39/73, IL-10, or the release of ROS [127]. MDSC accumulate within tissues and tumors during aging, though there is no evidence that they become more immunosuppressive [128,129]. Yet, the expression of p16 and p21 in MDSC was shown to stimulate the expression of the CX3CR1 chemokine receptor, which led to enhancing chemotaxis and the recruitment of MDSC into tumors [130]. The recruitment of MDSC expressing p16 is consistent with a previous report describing the presence of BM-derived p16-expressing cells in the TME [131]. However, it was noted that these MDSC are very unlikely to be in a state of cellular senescence, as they did not have other senescence markers [130]. Whether MDSC from aged mice express higher levels of p16 was not evaluated. 

In the same lines, increased tumor growth in older mice (2 vs. 12 months old) was associated with the accumulation of MDSC [132]. Mechanistically, it was shown that MDSC within the tumors of older mice had higher levels of Arginase1, a known inhibitor of proliferation. Others have reported a significant increase in the overall percentage of MDSC present in the bone marrow and spleens of aged compared to young animals [133]. However, the functions of these cells appeared not to be compromised in aged mice. The depletion of Tregs efficiently delays B16 tumor growth in young, but not in aged, mice, an effect that was mediated by an increase in the recruitment of MDSC [134]. However, the mechanistic link between Tregs and the recruitment of MDSC was not studied. Interestingly, an increased level of circulating MDSC was observed in melanoma patients and associated with a resistance to anti-CTLA-4 therapy [135]. In breast cancer patients, the level of circulating MDSC also correlated with the clinical stage and chemotherapy treatment efficacy [136]. Overall, it appears that phenotypic changes and an increase in the number of MDSC in aged mice may compromise the efficacy of ICI.

### 4.7. Tumor Growth Rate and Aging

An increase in the baseline tumor size is associated with a resistance to ICI and represents a negative prognostic factor for lung cancer and melanoma [137,138]. As discussed above, considering the overall age-related immunosuppression, it would be expected for tumors to grow faster in aged recipients. However, the literature is showing that the tumor growth rate in young vs. aged mice is largely divergent. For one, B16 melanoma, Engelbreth-Holm-Swarm, and BSC9AJ2 tumors were shown to grow faster in young vs. old mice [64,139,140,141]. Another study confirmed a delayed tumor growth in older compared to younger mice using several tumor cell lines (MCA-38, B16, and 4T1) [142]. In this study, an increased CD8 T-cell infiltration in MCA-38 tumors was attributed to an overexpression of the integrin VLA-4 (CD49d) and considered as a possible mechanism for reduced tumor growth. Finally, the growth of Met1, but not 4T1, tumors (and yet, two mammary cell line tumors) were shown to be delayed in older mice [91]. Several other studies demonstrated no effect of aging on tumor growth. For examples, the growth rates of B16 and TRAMP-C2 melanoma tumors were not reduced in old mice [134,140]. Nonetheless, another study showed that the TRAMP-C2 cell line grows faster in older than in younger mice, an effect that may be explained by an elevated number of TAMs and a higher extracellular matrix remodeling capacity [143]. Finally, two studies showed that B16 and TS/A tumor cells injected in mice tend to growth faster with age [58,132]. Furthermore, the growth of AE17 mesothelioma tumors was increased in aged mice, an effect associated with an increase in tumor-infiltrating macrophages [113]. The reasons for such discrepancies are not clear. Variations in the number of cells injected, the site of injection, and the age of mice, together with the in vitro genetic drifting of tumor cell lines over the years, may account for it (Table 1). Overall, these results suggest that the age of the host plays an important role in the tumor growth rate. We speculate that reduced growth rates in older patients may be a confounding effect, as it compensates for the reduced immune cell fitness in the overall impact of age in the success of immunotherapy treatments.

## 5. Therapeutic Possibilities to Enhance Immunotherapy in Aged Patients

### 5.1. Senescence Clearance

The administration of senolytic drugs is a promising strategy to prevent age-related pathologies. Two of the most tested senolytic treatments, D + Q and ABT263 (navitoclax), were, in fact, initially evaluated for their ability to kill tumor cells [144,145,146]. ABT263 has been in clinical trials, mostly for blood cancers, where it showed a limited effectiveness and severe treatment-induced thrombocytopenia [147,148]. This led to the development of the analog venetoclax, which limits the occurrence of thrombocytopenia but is unfortunately ineffective as a senolytic drug [149]. Nonetheless, the repurposing of these drugs (perhaps at lower doses to limit the major side effects) or using one of the many novel senolytic drugs currently in development could, by eliminating senescent cells, reinvigorate immune cells and enhance the efficacy of immunotherapies. Few studies so far have explored the possibility of combining senolytic drugs with classical chemotherapy treatments. For one, the combination of D + Q with doxorubicin was shown mostly ineffective at further reducing tumor growths, as D + Q treatments were unable to eliminate doxorubicin-induced senescent cells [150]. However, it is likely that D + Q was injected too early (2 days) after the injection of doxorubicin, leaving no time for the induction of the senescent phenotype. In another study, ABT263 was used in combination with poly (ADP-ribose) polymerase 1 inhibitors (PARPi). This proved to be effective at reducing the growth of ovarian and breast tumors [151]. Hypothetically, we believe that the beneficial effect of a senolytic treatment in aged patients could help at different levels (Figure 1). For one, the elimination of treatment-induced senescent cells slows down tumor growths by preventing the direct mitogenic effect of the SASP on residual tumor cells [76]. Second, this effect could be indirect through the elimination of senescent cells from the TME, which could favor the tumor immune response. The reversal of myeloid cells skewing, which was demonstrated with ABT263, is another mechanism by which senolytics may improve immunotherapies [40] (Figure 1).

To our knowledge, the preclinical evaluation of the combination of a senolytic drug with an ICI is lacking. Yet, other strategies looking to improve ICI have been recently tested and are discussed below.

### 5.2. Cytokine Blocking

Different preclinical studies have tried blocking proinflammatory cytokines, hoping to improve the efficacy of ICI [152]. Interestingly, most of the targeted cytokines overlapped with the SASP, supporting the hypothesis that these approaches can help suppress tumor growth in older patients [124].

#### 5.2.1. IL-6 Inhibition

Very recently, a strategy focusing on blocking IL-6 to decrease the recruitment and the immunosuppressive ability of MDSC was tested [153]. However, this treatment failed to enhance the efficacy of PD-1-blocking antibodies in melanoma. The authors hypothesized that primary IL-6 signaling is necessary to activate immune cells. Nevertheless, this strategy may be more efficient in older patients, where the circulating level of IL-6 is higher [154].

#### 5.2.2. IL-10 Inhibition

IL-10 is an important immunosuppressive cytokine secreted by MDSC [126] and T cells/Tregs isolated from aged mice [107,108]. Other than inhibiting T-cell activation, IL-10 can boost the expression of PD-1 on tumor-infiltrated DCs and make tumor cells more resistant [155]. Interestingly, the blockade of IL-10 by using a neutralizing antibody decreases the accumulation of MDSC within the TME and enhances the efficacy of the anti-PD-1 therapy [155]. We speculate that blocking IL-10 pathways in association with ICI could be beneficial in aged mice, as they exhibit an increased systemic level of IL-10 [108].

#### 5.2.3. CCL-2 Inhibition

C-C Motif Chemokine Ligand 2 (CCL2, or MCP-1 in mice) is a chemokine notably responsible for the recruitment of MDSC in the TME [126]. Hence, blocking the CCL2-CCR2 (CCL2 receptor) axis represents an interesting strategy to enhance the efficacy of immunotherapies. In this context, the median survival was found to be increased in a mouse model of glioma using a combination of CCR2 antagonist (CCX872) and PD-1-blocking antibodies [156]. In addition, a similar effect was observed with the same combination tested on several other tumor cell lines (bladder, melanoma pulmonary metastasis, and mammary tumors) [157]. As the level of CCL2/MCP-1 constantly increases during aging, blocking this pathway may alleviate the tumor resistance and improve the ICI efficacy [158].

#### 5.2.4. IFN Pathway Modulation

Sustained IFN type I exposure was shown to lead to the resistance of B16 melanoma tumors to ICI therapy through a mechanism involving the overexpression of several immune checkpoints (CD86, TNFRSF14, PDL1, and LGALS9) [159,160]. In the same context, IFN type II was shown to increase the resistance of tumor cells to the cytotoxic action of CD8 T cells by enhancing the expression of SERPINB9 [161]. Nonetheless, the literature concerning the role of IFN type I is widely divergent. In fact, the secretion of IFN type I by infiltrated T cells within the TME elicits the apoptosis of MDSC and is associated with successful anti-PD-1 therapy [162]. Additionally, in aged mice (12 months), a treatment with a STING agonist (5,6-dimethylxanthenone-4-acetic acid (DMXAA)), used to enhance IFN type I signaling, can increase the efficacy of anti-PD-L1 and anti-CTLA-4 therapies [91]. Interestingly, the responses to IFN type I signaling are reduced in T cells isolated from older humans (65–85 vs. 20–35 years) [163]. Overall, it seems that short-term, but not chronic activation, of the IFN pathway in aged patients could be beneficial to improve the efficacy of immunotherapy.

#### 5.2.5. TNFα Inhibition

Blocking TNFα is another promising strategy that was shown to enhance anti-PD-1 therapy in a mouse model of melanoma [164]. A phase 1 clinical trial based on this approach is currently active (TICIMEL-NCT03293784). In support of this approach, the level of circulating TNFα was found to be enhanced in aged mice and in a premature senescence model induced by mitochondrial dysfunction in T cells [165].

## 6. Conclusions

In summary, the impact of senescence on many age-related pathologies, including cancer, is of growing interest. We believe that the age of patients at the time of treatment is poorly taken into consideration in preclinical research, despite an increasing amount of evidence highlighting its important role. It is clear that aging and senescence deeply modify the TME by favoring the accumulation of many types of immunosuppressive cells that will necessarily have an impact by making tumor cells more resistant and more prone to evade the immune system. As ICI fails in a substantial number of patients, identifying the mechanisms responsible for this failure is necessary to improve the efficacy of treatments. In the future, we believe it is crucial to include aged subjects in preclinical studies and, if possible, younger patients in clinical studies to thoroughly assess the impact of aging and senescence on the success of cancer treatments. The development of new senolytic treatments able to efficiently clear all types of senescent cells could represent a remarkably interesting strategy to improve immunotherapies.

## Figures and Tables

**Figure 1 cells-10-01568-f001:**
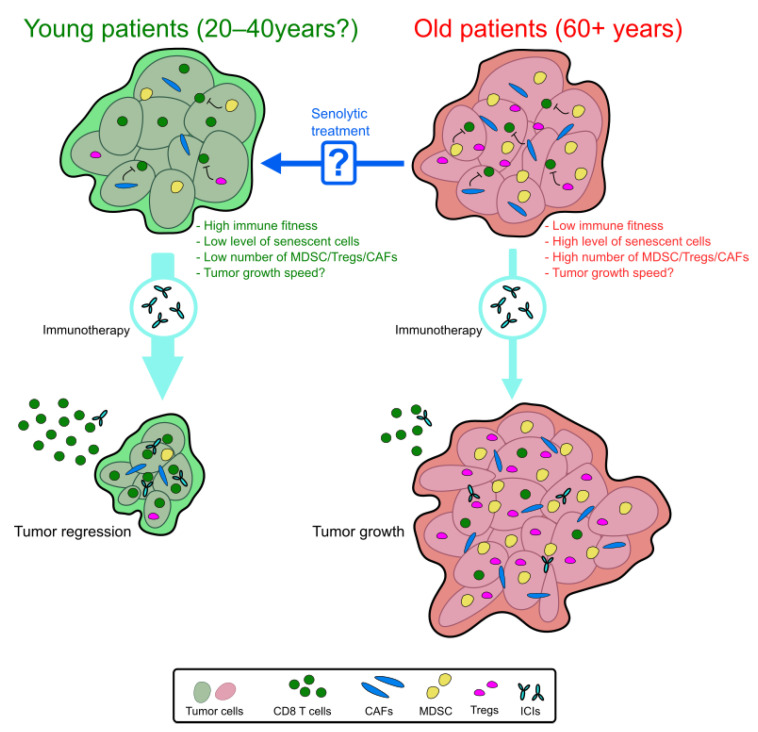
Predictive model of the immunotherapy efficacy in young and old patients. In old patients, we expected a loss of immunotherapy efficacy. Within the TME, the decrease in immune cell fitness with the enrichment of senescent cells and immunosuppressive cells (MDSC, Tregs, and CAFs) can be responsible for this resistance. Pretreatment with a senolytic drug could reverse this immunosuppressive context and enhance the immunotherapy efficacy in aged patients.

**Table 1 cells-10-01568-t001:** Comparison of the tumor growth rate in young and aged mice.

Cell Lines	Young Group Age	Old Group Age	Mouse Strain	Number of Cells Injected	Injection Site	Tumor Growth in Old vs Young	References
B16/F10	2.5–6 months	19–26 months	C57/BL/6	500,000	Intradermal	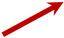	Padrón et al., 2018 [58]
BSC9AJ2	8–10 weeks	10 months	C57BL/6	100,000	Intradermal	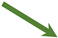	Kugel et al., 2018 [64]
4T1	8–10 weeks	>12 months	Balb/C	100,000	Mamarry fatpad	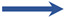	Sceneay et al., 2019 [91]
Met1	FVB	200,000	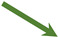
AE17	3 months	20–24 months	C57BL/6	500,000	Subcutaneous	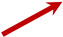	Duong et al., 2018 [113]
TS/A	2 months	12 months	BXD12	150,000	Subcutaneous	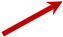	Grizzle et al., 2007 [132]
B16/F10	2–6 months	22–26 months	C57BL/6	250,000	Subcutaneous	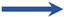	Hurez et al., 2012 [134]
MC38	1,000,000
B16/F10	3 months	24 months	C57BL/6	100,000	Subcutaneous	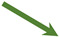	Ershler et al., 1984 [139]
B16/F1	Intravenous
B16/F10	4 months	20 months	C57BL/6	1,000,000	Subcutaneous	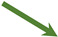	Reed et al., 2006 [140]
TRAMP-C2	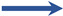
MC38	3–4 months	12–15 months	C57BL/6	1,000,000	Subcutaneous	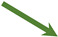	Oh et al., 2018 [142]
B16/F10
4T1	Balb/C
TRAMP-C2	4 months	20–24 months	C57BL/6	500,000	Dorsolateral prostate	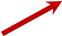	Bianchi-Frias et al., 2019 [143]

## Data Availability

All the data generated or analyzed during this study are included in this article.

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
