# Peer review of "Senescence and Aging: Does It Impact Cancer Immunotherapies?"

_cells, 2021, doi:10.3390/cells10071568_

Round 1

Reviewer 1 Report

The review “Senescence and aging: Does it impact cancer immunotherapies?“ by Damien Maggiorani and Christian Beauséjour is very timely and addresses an important aspect of why cancer immunotherapies may fail in cancer patients. The review provides very interesting aspects on the topic. I suggest minor revisions regarding the points below:

Page 2, line 69-78: 2 References could be included as additional examples, e.g. that senolytic CAR T cells can reverse senescence-associated pathologies (https://doi.org/10.1038/s41586-020-2403-9 would also fit to chapter 5.1) and that ageing hallmarks exhibit organ-specific temporal signatures (https://doi.org/10.1038/s41586-020-2499-y). Another important and recent review by Campisi et al. “From discoveries in ageing research to therapeutics for healthy ageing” could be included too (https://doi.org/10.1038/s41586-019-1365-2).

Chapter 4.3:

The authors should include a short paragraph discussing the decline of naïve T cell production in old age due to thymic involution. The dramatic decline in the production of new, naïve T cells leads to a skewed TCR repertoire among the remaining naïve T cells, which decreases the capacity of naïve T cells in older adults to respond to neoantigens (tumor neoantigens but also observed after vaccination with neoantigens).

Reviews by Goronzy and Weynand (https://pubmed.ncbi.nlm.nih.gov/31186548/) and Akbar and Henson (https://pubmed.ncbi.nlm.nih.gov/21436838/) should be included as they discuss the important topics of T cell aging as well as senescence vs. exhaustion

An interesting report by Martinez-Jimenez et al., Science, 2017 (https://pubmed.ncbi.nlm.nih.gov/28360329/), which shows that aging increases cell-to-cell transcriptional variability upon immune stimulation could be included and briefly discussed.

There are still numerous errors regarding grammar and sentence structure that need to be corrected to improve readability.

Author Response

Response to reviewer 1:

Comment: Page 2, line 69-78: 2 References could be included as additional examples, e.g. that senolytic CAR T cells can reverse senescence-associated pathologies (https://doi.org/10.1038/s41586-020-2403-9 would also fit to chapter 5.1) and

Answer: Yes, we have added the work by Scott Lowe on CAR T cells (now ref 45, page2 , line 79).

Comment: ageing hallmarks exhibit organ-specific temporal signatures

Answer: Yes, this is indeed an important manuscript.  We are now citing and briefly describing this manuscript on page 2 line 50 (new references 21 and 22).

Comment: A recent review by Campisi et al. “From discoveries in ageing research to therapeutics for healthy ageing” could be included too

Answer: The manuscript is now cited (new ref 6) page 1 line 24.

Comment: The authors should include a short paragraph discussing the decline of naïve T cell production in old age due to thymic involution. The dramatic decline in the production of new, naïve T cells leads to a skewed TCR repertoire among the remaining naïve T cells, which decreases the capacity of naïve T cells in older adults to respond to neoantigens (tumor neoantigens but also observed after vaccination with neoantigens).

Answer: A short paragraph was added in Section D – Modification of leukocytes with age on 4 line 178. New references citing this work were added (ref 83-86).

Comment: Reviews by Goronzy and Weynand (https://pubmed.ncbi.nlm.nih.gov/31186548/) and Akbar and Henson (https://pubmed.ncbi.nlm.nih.gov/21436838/) should be included as they discuss the important topics of T cell aging as well as senescence vs. exhaustion

Answer: These reviews were added. (ref 84 and 93).

Comment: An interesting report by Martinez-Jimenez et al., Science, 2017 (https://pubmed.ncbi.nlm.nih.gov/28360329/), which shows that aging increases cell-to-cell transcriptional variability upon immune stimulation could be included and briefly discussed.

Answer: This manuscript is now briefly discussed on page 5 line 196. 

Comment: There are still numerous errors regarding grammar and sentence structure that need to be corrected to improve readability.

Answer: The manuscript grammar was improved to the best of our capabilities.

Reviewer 2 Report

Overview and general recommendation: The present manuscript is a review paper about cellular senescence and aging and how senescence and aging impact cancer immunotherapies. Immuno-senescence is characterized by the loss of function and proliferation potential of immune cells. Immune checkpoint inhibitors (ICIs) have made an indelible mark in the field of cancer immunotherapy. However, the importance of aging and senescence on the efficacy of ICIs remains under-evaluated in pre-clinical studies. Therefore, it is essential to know whether cellular senescence affects the effectiveness of anti-cancer treatment and, in particular, the ability of cancer cells to evade the immune system. This topic is timely. The authors reviewed senescence and aging in the immunotherapy era, the possible age-associated mechanisms that affect the efficacy of immunotherapy systemically, the therapeutic possibilities to enhance immunotherapy in aged patients systemically. This manuscript is well written, and I believe it will be valuable to Oncologists, Immunologists, and Cancer-related aging Researchers.

Minor Comments:

  1. Page 3, lines 97-100. The authors mentioned that “patients over 60 years old, the equivalent of a 15-18 months old mouse……around 3 or 6 months old mice, an equivalent of 20-35 years old patients”. These statements need references. As these statements are much different from what I know, the ages equivalent between rat and human.
  2. Page 4, line 182. “…, work by Sceneay and al…”.Should be “work by Scenery et al.”
  3. Page 5, line 197. “At the opposite”,…Should be “On the opposite”.
  4. Page 5, line 199. “in a patients over 60 years old”. Should be “in patients over 60 years old”.

Author Response

Response to reviewer 2:

Comment: Page 3, lines 97-100. The authors mentioned that “patients over 60 years old, the equivalent of a 15-18 months old mouse……around 3 or 6 months old mice, an equivalent of 20-35 years old patients”. These statements need references.

Answer: A reference was added (ref 57) and 20-35 changed to 20 years old. 

Comment:       Page 4, line 182. “…, work by Sceneay and al…”.Should be “work by Scenery et al.”

Answer: Modification was made. Thank you

Comment: Page 5, line 197. “At the opposite”,…Should be “On the opposite”. Page 5, line 199. “in a patients over 60 years old”. Should be “in patients over 60 years

Answer: Both modifications were made.  Thank you.

We hope we have satisfactorily answered the reviewer’s comments.